# Spatial Variation of PM_2.5_ Indoors and Outdoors: Results from 261 Regulatory Monitors Compared to 14,000 Low-Cost Monitors in Three Western States over 4.7 Years

**DOI:** 10.3390/s23094387

**Published:** 2023-04-29

**Authors:** Lance Wallace, Tongke Zhao

**Affiliations:** 1Independent Researcher, 428 Woodley Way, Santa Rosa, CA 95409, USA; 2Independent Researcher, Milpitas, CA 95035, USA

**Keywords:** spatial variation, PM_2.5_, PurpleAir, PM_2.5__alt, coefficient of divergence, FEM, FRM, low-cost monitors

## Abstract

Spatial variation of indoor and outdoor PM_2.5_ within three states for a five-year period is studied using regulatory and low-cost PurpleAir monitors. Most of these data were collected in an earlier study (Wallace et al., 2022 *Indoor Air* 32:13105) investigating the relative contribution of indoor-generated and outdoor-infiltrated particles to indoor exposures. About 260 regulatory monitors and ~10,000 outdoor and ~4000 indoor PurpleAir monitors are included. Daily mean PM_2.5_ concentrations, correlations, and coefficients of divergence (COD) are calculated for pairs of monitors at distances ranging from 0 (collocated) to 200 km. We use a transparent and reproducible open algorithm that avoids the use of the proprietary algorithms provided by the manufacturer of the sensors in PurpleAir PA-I and PA-II monitors. The algorithm is available on the PurpleAir API website under the name “PM_2.5__alt”. This algorithm is validated using several hundred pairs of regulatory and PurpleAir monitors separated by up to 0.5 km. The PM_2.5_ spatial variation outdoors is homogeneous with high correlations to at least 10 km, as shown by the COD index under 0.2. There is also a steady improvement in outdoor PM_2.5_ concentrations with increasing distance from the regulatory monitors. The spatial variation of indoor PM_2.5_ is not homogeneous even at distances < 100 m. There is good agreement between PurpleAir outdoor monitors located <100 m apart and collocated Federal Equivalent Methods (FEM).

## 1. Introduction

The spatial variation of airborne fine particles (e.g., PM_2.5_) has long been an interest of environmental regulatory agencies. This interest is due to the sparse nature of the monitoring networks, with monitors separated by scores or hundreds of miles. If the particle levels in the region between monitors are spatially homogeneous, then the monitoring networks would provide reasonable evidence of outdoor concentrations throughout the network area.

However, the nature of environmental regulation requires that the responsible agencies make some effort to include those areas with the highest expected annual particle concentrations (for example, two regulatory sites in southern California are located on both sides of the expressway with the highest traffic flow, thus ensuring that at least one monitor will be downwind.) A somewhat competing priority is to estimate the general exposure of the population, which often leads to placing monitors in areas with the highest density of population. Both of these priorities tend to produce higher concentrations at the regulatory monitoring sites than elsewhere in the urban area. 

A second group deeply interested in spatial variation is health scientists and epidemiologists. They face the same problem of sparse outdoor networks with a need to interpolate or attribute outdoor concentrations to homes or areas with few nearby measurements. Epidemiologists create exposure-response curves to quantify the effects of PM_2.5_. They often have data on the health status of many thousands of persons living in a large area but typically have only a few outdoor sites to estimate exposure. If they could be sure that those outdoor sites are highly correlated, they would have a firmer basis for their models. This is one reason why so many papers are written on the spatial variation of outdoor sites. Our efforts provide extremely detailed results on the spatial variation of those outdoor sites, and, therefore, could be of use to epidemiologists, at least in these three states. In addition, we include estimates of indoor PM_2.5_, using data collected over 4.7 years in three Western states [1]. 

Many articles on spatial variation have been written by both regulatory agency staff and epidemiologists. A nationwide intercomparison of regulatory sites in 27 urban areas in the United States was carried out for the 1999–2000 years [2]. In general, correlations were relatively high, but the authors argued that estimated concentrations could be quite different but still have high correlations. They proposed using coefficients of divergence (COD) as a complementary measure since the COD is a direct measure of difference in concentration:(1)$$\mathrm{COD}=\sqrt{\frac{1}{n}\overset{n}{\underset{1}{\sum }}{\left(\frac{x_{i1}-x_{i2}\;}{x_{i1}+x_{i2}}\right)}^{2}}$$
where *n* is the number of joint measurements made at sites 1 and 2, and the fraction following the summation sign is the difference divided by the sum of the two measurements. The COD is zero if there is perfect agreement between the two datasets. It rises to 1 if, for example, one measurement is zero and the corresponding measurement is not zero. The authors reported that both the correlation coefficients and the COD varied more widely than expected since the measurements made at regulatory sites are considered among the best that can be done, particularly the gravimetric Federal Reference Method (FRM) but also the Federal Equivalent Method (FEM). In the West Coast states, about 33 monitoring sites were included in 6 urban areas, including Seattle, Portland, San Francisco, Los Angeles, Riverside-San Bernadino, and San Diego. With the exception of three outliers, correlation coefficients ranged from 0.57 to 0.99. The COD ranged from 0.11 to 0.26. Since the ranges of both these coefficients were large, the authors concluded that the degree of heterogeneity varied quite widely and could cause exposure misclassification if homogeneity were assumed. 

The natural question arises of what COD value would indicate a homogeneous relationship. That is, how much error should we allow to still assume that a measurement at one site will be a reasonable approximation to a measurement at another site? One study suggested that a 20% error (COD = 0.2) might be a reasonable maximum for considering two sites or methods to be homogenous [3]. A thoughtful major review of 40 studies of PM_2.5_ or PM_10_ concentrations measured by regulatory monitors in multiple sites (mostly intraurban) adopted this cutoff of 0.2, finding that 16 studies could be considered homogeneous and 24 heterogeneous [4]. For example, three studies found Philadelphia to be homogenous with respect to PM_2.5_, but three other studies found Los Angeles to be heterogenous. A later complementary review added about 20 studies [5]. 

Low-cost particle monitors are increasingly being used to measure outdoor air quality. In some areas, they are clustered in such quantities that they can be used to estimate the spatial and temporal variation of PM_2.5_ with increased resolution compared to studies using mainly regulatory monitors. Multiple studies have been carried out using many different low-cost monitors [6,7,8,9,10,11,12,13,14,15,16,17,18,19,20,21,22,23,24]. A useful source of information for many of these monitors is the AQ-SPEC program providing laboratory and field comparisons for scores of monitors produced by different manufacturers [25]. 

One of the better-performing monitors in the AQ-SPEC record was the PurpleAir PA-II monitor containing two Plantower PMS 5003 sensors (https://www2.purpleair.com/, accessed on 27 April 2023, https://www.plantower.com/en/, accessed on 27 April 2023). PurpleAir has one of the largest networks of monitors operating around the world and maintains a publicly available database for all monitors. The inclusion of two identical but independent sensors provides a quality control opportunity for every measurement. The existence of the PurpleAir and other low-cost monitor networks has made it possible for the first time to use actual measured data with extremely detailed resolution to determine spatial variation.

In this paper, we first compare the correlations and CODs between PurpleAir monitors and regulatory sites using Federal Equivalent Methods (FEM) or Federal Reference Methods (FRM). The correlations and COD results provide an indication of the quality of the PurpleAir measurements. Second, the correlations and COD results for pairs of PurpleAir indoor monitors and also for pairs of PurpleAir outdoor monitors are calculated. This provides an indication of the replicability of PurpleAir measurements within each type of environment. Finally, the correlations and CODs of pairs of PurpleAir monitors at a range of distances from 50 m to 50 km show the rate of decline of the correlations (or rate of increase for the COD values) with the increasing distance apart. 

To our knowledge, no long-term (months to years) large-scale (hundreds to thousands of homes) studies of concentrations, correlations, and CODs between indoor sites at varying distances apart has been undertaken. This is due mainly to the fact that these long-term data on indoor particles did not exist until the development of small quiet low-cost monitors that can measure indoor levels over months and years. This lack of indoor data has forced epidemiologists to study only exposure to particles of ambient origin. Their assumption is that nearby homes will all experience about the same exposure to particles of ambient origin. This assumption can now be directly checked using measured PM_2.5_ data from some thousands of PurpleAir monitors. A second assumption (usually unvoiced) is that indoor-generated particles have no effect on human health. Only with this assumption can health effects due to actual exposure to particles from all sources be related to ambient particles alone. However, health effects can be expected from some indoor-generated particles such as those created by smoking (tobacco, marijuana) and high-temperature cooking, particularly using biomass (wood, dung) as fuel.

Although we cannot know the conditions in all homes, or even in any homes, we can state that this is a complete census (not a sample) of all 4000 homes with indoor monitors in three states. We also know that for many homes, the monitoring period was at least one year, so we have data on all seasons for 4.7 years between 2017 and 8 September 2021. These years included wildfires in some locations, so our data also include periods of extremely high outdoor concentrations (and, therefore, in many cases, correspondingly high indoor concentrations due to infiltration). We also include, in some cases, extremely high indoor concentrations without high outdoor concentrations, which would indicate a high level of indoor-generated particles. There are also cases of extremely low indoor concentrations, possibly attributable to the use of air cleaners. This could be a fruitful area for further studies, but it is beyond our scope here to look at spatial correlations.

## 2. Methods and Materials

A previous study collected publicly available particle number concentrations from ~10,000 outdoor and 4000 indoor PurpleAir monitors in the states of Washington, Oregon, and California, covering the 4.7-year period from 1 January 2017 to 8 September 2021 [1]. There were 9910 outdoor PurpleAir PA-II monitors with two PMS 5003 sensors. The indoor monitors consisted of 1178 PA-II monitors and 3500 PA-I monitors with a single PMS 1003 sensor. The locations of the 4678 indoor monitors are provided (Figure 1). 

In the present study, we downloaded all daily mean PM_2.5_ concentrations for 261 US EPA regulatory monitors in the three West Coast states for the 5-year period from 2017 through 2021 (https://aqs.epa.gov/aqsweb/airdata/download_files.html#Daily, accessed on 27 April 2023) These instruments are located in 135 unique sites in multiple cities located in the three states of Washington, Oregon, and California. They are under the authority of the US EPA, located in Washington, DC and Research Triangle Park, NC. Locations of the regulatory monitors are shown (Figure 2). 

For the PurpleAir sites, PM_2.5_ concentrations were calculated using an improved algorithm based on particle numbers in three size categories as reported by the PurpleAir monitors. This algorithm is completely independent of the two proprietary algorithms provided by Plantower and has been shown to have reduced bias, improved precision, and a lower Limit of Detection (LOD) [1,8,26,27,28]. The algorithm is called ALT-CF3 and is available on the PurpleAir main page as one of 5 “conversion factors” that can be chosen instead of the Plantower proprietary algorithms. The algorithm is also available on the PurpleAir API site, where it is called “PM_2.5__alt” (https://api.purpleair.com/, accessed on 27 April 2023). Briefly, the algorithm uses the particle numbers reported by the Plantower sensors to calculate PM_2.5_. It is assumed that all particles are spherical and have diameters equal to the geometric mean of the boundaries of their size categories. The three size categories used are 0.3–0.5, 0.5–1, and 1–2.5 µm. An arbitrary density (in this case, that of water) is assumed. The resulting PM_2.5_ mass estimate is then calibrated by comparison of nearby PurpleAir monitors to regulatory monitors. The first estimate of the calibration factor employed 33 PurpleAir monitors within 0.5 km of regulatory monitors in the state of California [26]. That study found a CF of 3, which led to naming the algorithm ALT-CF3. A later study using both PA-I and PA-II monitors in the three West Coast states found a calibration factor of 3.4 for both the Plantower 1003 and 5003 monitors [27].

Software employed includes Python 3.11.1 (Wilmington, DE, USA); Statistica v.11 (Statsoft. Tulsa, OK, USA); and Excel 2013 (Redmond, Washington, DC, USA). 

### 2.1. Comparisons with Regulatory Sites

All PurpleAir outdoor sites were matched with FEM/FRM regulatory sites within 50 km distance. The 50 km upper limit was selected based on our initial findings of high correlations at 10 km. We, therefore, looked at larger distances of 20 and 50 km. At least 30 days with valid daily averages for each pair of sites were required. We chose 8 nonoverlapping distances from 0–100, 100–500, up to 20–50 km. For each distance range, the PurpleAir sites were regressed on the FEM/FRM sites and mean and median values of PM_2.5_, Pearson and Spearman correlation coefficients, and COD values were determined.

### 2.2. Intercomparisons of PurpleAir Outdoor-Outdoor and Indoor-Indoor Pairs

All pairs of PurpleAir outdoor sites and separately all pairs of indoor sites s within 50 km distance apart were examined, again with a requirement for at least 30 days of joint valid daily means. For each distance selected, we considered all possible pairs of monitors separated by less than that distance. The number of pairs of N monitors is N(N − 1)/2. For example, the total number of pairs of outdoor sites within 50 km of each other exceeded 49 million. Because of the very large numbers of paired sites >2 km apart, random samples of 10,000 pairs of sites between 2, 5, 10, 20, and 50 km apart were selected. A second independent random sample was run, and the results were compared to determine whether 10,000 was a proper number to ensure the stability of the results.

### 2.3. Intercomparison of Regulatory Sites

The regulatory sites operate two types of monitors employing gravimetric Federal Reference Methods (FRM) and continuous Federal Equivalent Methods (FEM). In theory, the FEM monitors should be equivalent to the FRM monitors. For the 68 pairs of collocated monitors, we compared FRM-FRM, FEM-FEM, and FEM-FRM pairs to test whether the FEM monitors are capable of matching the gravimetric FRM monitors under field conditions over the past 5 years.

## 3. Results

### 3.1. Comparisons with Regulatory Sites

There were 82,562 pairs of PurpleAir and regulatory sites within 50 km apart (Table 1). The mean number of days per matched pair was 304 (about 10 months). A total of 25,113,076 matched days were considered. For distances apart up to 0.5 km, the PM_2.5_ means reported by the Federal regulatory and PurpleAir monitors using the ALT-CF3 algorithm agreed to within 2%. The Pearson correlation coefficient declined slowly from 0.91 at 100 m distance to 0.81 at 20 km distance. The coefficient of divergence increased from 0.24 at 100 m to 0.32 at 50 km.

As the distance between the PurpleAir and regulatory monitors increased, the PM_2.5_ concentration declined (i.e., the air quality improved) (Figure 3). The improvement was already noticeable at distances from 0.5 to 1 km. The difference was significant for all distances >1 km from the regulatory monitor.

### 3.2. Intercomparison of Indoor-Indoor and Outdoor-Outdoor PurpleAir Pairs

Pairs of outdoor PurpleAir sites and separately pairs of indoor sites at nine nonoverlapping rings of increasing distances apart from 0 to 50 km were selected (Table 2). All pairs were included up to 1 km for both datasets. For larger distances, random samples of 10,000 pairs were carried out. The second independent random sample agreed with the first to within 0.1–1%, and we concluded the number of 10,000 was sufficient to ensure stability. For each distance range, about 7500 pairs of outdoor and indoor sites met the requirement of at least 30 days of joint measurements by each monitor. The outdoor sites showed mean and median correlations from 0.97 to 0.998 for distances apart less than 50 m and were still ranging between 0.88 and 0.95 up to 10 km distance apart. The COD estimates were <0.2 (suggesting homogeneity) out to 10 km. 

By contrast, the indoor sites had only moderate Spearman and Pearson correlations at the smallest distance apart of 50 m. The COD mean and median values at all distances never approached the 0.20 value associated with homogeneity. 

The mean CODs for the outdoor monitors are contrasted with those for the indoor monitors (Figure 4).

### 3.3. Intercomparison of Regulatory (FRM/FEM) Sites

There were 175 regulatory sites and 261 unique regulatory monitors operating between 2017 and 2021 in the three West Coast states. At each distance category >0 km, ~600 days of joint monitoring were observed for a total of 3,710,522 days. Two more rings were added (50–100 km and 100–200 km) in recognition that these measurements were expected to be the best available and might be correlated over longer distances than the 50 km maximum chosen for the PurpleAir monitors. Mean Pearson correlations ranged from 0.93 for collocated monitors down to 0.77 (still moderately correlated) at 50 km (Table 3). However, at 100 and 200 km, the correlations dropped to 0.63 and 0.52 (R^2^ only 41% and 27%), so 50 km may be an upper limit for reasonable correlations to be found. The mean and median CODs were very similar, ranging from 0.13 or 0.14 for the collocated monitors to 0.18 at 20 km, suggesting that 20 km might be considered a cutoff limit for homogeneity of PM_2.5_ as measured by the regulatory monitors. 

### 3.4. Comparison of Collocated FEM and FRM Monitors

There were 68 pairs of collocated instruments adding up to 25,147 days with matched measurements. Correlations and CODs of collocated instruments using matched methods (FRM-FRM, FRM-FEM, and FEM-FEM) are provided (Table 4). The better performance of the gravimetric method is indicated by the R^2^ values of 0.96 and 0.99 for the matched FRM monitors compared to 0.82 and 0.88 for the matched FEM monitors. The COD results also favor the FRM-FRM pairs (<0.1) over the FEM-FEM pairs (<0.15), but even the latter meet the requirement for homogeneity (COD < 0.2).

## 4. Discussion

### 4.1. Comparisons of PurpleAir with Regulatory Sites

There was excellent agreement (to within 2%) between regulatory and PurpleAir monitor estimates of PM_2.5_ for distances of 0–0.5 km. This result confirms the most recent calibration factor of 3.4 for both PMS 1003 and PMS 5003 sensors using the ALT-CF3 algorithm [27]. The PurpleAir mean PM_2.5_ decreases somewhat with increasing distance from the regulatory sites, reaching about 13% lower than the regulatory monitors. This is expected since most residences are located away from the center city and other areas with expected higher emissions (e.g., busy roadways). The higher incomes associated with homeowners using PurpleAir monitors would also allow living in areas with better environmental surroundings [15]. However, the increasing difference between regulatory and PurpleAir measurements with distance directly affects COD values, even though we believe both measurements are correct. The COD values may be misleadingly high if the monitors of one type are located in areas that have different concentrations than those with monitors of the second type. 

### 4.2. Intercomparisons of PurpleAir Outdoor Sites

PurpleAir outdoor sites showed very high agreement, with median Pearson correlations at 100 m at an extraordinarily high level of 0.998, falling only to 0.964 at a 20 km distance. This indicates a high degree of dependability and replicability of the sensors. Although this dataset does not include a census of all available matched pairs, two independent random samples were taken, with variations of mean concentrations and correlations at the 0.1–1% level, therefore, the findings seem robust.

The period covered included all major fires in a 4.7-year period with very high PM_2.5_ concentrations, as well as periods with widespread rain and resulting PM_2.5_ concentrations near zero. During the 2017 through 2021 period in the full tristate area, weekly mean PM_2.5_ concentrations never rose above 13.5 µg/m^3^ in the year 2019, whereas all other years had maximum weekly values exceeding 20 µg/m^3^ and one week reached 60 µg/m^3^. The wildfire “season” appeared to run from August through December, with 19 of the 20 highest weekly values occurring in those months. The lowest weekly values observed were just above 2 µg/m^3^. Of course, if we further narrowed it down to, say, 1-day means per county, very much higher (and even lower) values were observed. Since there were typically 200–300 days of matched daily means, the results appear robust against extreme variations in PM_2.5_ concentrations. 

### 4.3. Intercomparisons of PurpleAir Indoor Sites

These measurements are the first, we believe, for multiple homes with typically hundreds of days of measurements. Indoor Pearson correlations were much lower (about half) than outdoor correlations at comparable distances apart. It has been shown [1] that indoor-generated particles contributed, on average, about half of the total indoor air concentration, so the reduction (also by about half) of the correlation coefficients clearly reflects the lack of our ability to predict indoor exposures by using only ambient measurements. Indoor-generated particles are often considered to be independent of outdoor concentrations [29], and these low correlations support that idea. 

The COD mean values for indoor particle pairs are sharply higher (0.40 to 0.43) than those for outdoor particle pairs (0.14 to 0.26). If we take 0.20 as the upper boundary for relative homogeneity, then the distance for homogeneity of outdoor particle concentrations would extend to about 10 km. However, for indoor concentrations, there is no homogeneity even within 1 km distance.

### 4.4. Intercomparison of Regulatory Sites

The 68 regulatory sites with collocated monitors showed excellent performance by the FRM monitor pairs (99% R^2^) but lower values for the FEM monitor pairs (82–88%). More than 100 3-year studies of collocated FRM and FEM monitors have been carried out by the EPA (https://www.epa.gov/outdoor-air-quality-data/pm25-continuous-monitor-comparability-assessments, accessed on 27 April 2023). A considerable fraction of these studies showed disagreement >20% between the methods. This field performance by FEM monitors suggests a need for improved quality assurance at EPA regulatory sites.

The mean and median correlations of the FEM-FEM collocated pairs (distance 0) ranged from 0.90 to 0.94. Although we had no collocated pairs of PurpleAir monitors, the 1000 monitors within 100 m of each other had correlations ranging from 0.96 to 0.99, better performance than the FEM monitor pairs. The mean and median COD values for the collocated FEM pairs (0.11 and 0.15) were also not quite as good as those for the PurpleAir monitors up to 100 m apart (0.07–0.13).

### 4.5. Future Research

The data collected in [1] and partially analyzed in this paper could be explored in more depth by interested investigators. For example, diurnal and seasonal variation, effects of wildfires on indoor and outdoor PM_2.5_ concentrations, and many other approaches could be undertaken using the 3.8 million daily averages of outdoor PurpleAir monitors together with the 1.3 million daily averages of indoor PurpleAir monitors. Using our findings on spatial variation of outdoor PM_2.5_ could contribute to network design, for example, just a few outdoor monitors might represent concentrations at a given large distance quite well.

## 5. Conclusions

For 356 PurpleAir sites within 0.5 km of regulatory sites, PM_2.5_ mean concentrations agreed with FEM/FRM measurements to within 2%. This result appears to validate the algorithm and calibration factor of 3.4 used here for all PurpleAir monitors [27]. 

Outdoor PM_2.5_ fell almost monotonically with increasing distance from the regulatory monitors, reaching 12–13% improvement at distances >10 km. This can be considered a yardstick for measuring quantitative changes in fine-particle pollution with distance from center-city locations.

Outdoor mean PM_2.5_ concentrations were highly correlated, with evidence of homogeneity (COD < 0.2) out to distances of 10 km. Indoor concentrations were poorly correlated and heterogenous (COD > 0.2) at all distances. Correlations were about half those for outdoor pairs of monitors, possibly reflecting the finding in a previous study [1] that indoor-generated particles contributed about half of the total potential indoor PM_2.5_ exposure. In particular, we find that even at the smallest distances apart of 0–50 m, the mean COD values exceed 0.3, well above the cutoff of 0.2 established in many studies to indicate reasonable agreement. We conclude, in the context of our study, that indoor PM_2.5_ exposures cannot be estimated quantitatively from outdoor measurements alone.

Finally, our analysis of 68 collocated regulatory monitors over 5 years shows high correlations and good COD values for FRM-FRM pairs, with somewhat lower correlations and slightly worse but still good (i.e., <0.2) COD values for collocated FEM-FEM pairs. However, a direct comparison with PurpleAir monitors at distances up to 100 m apart slightly favored the PurpleAir monitors, suggesting that they are at least comparable in accuracy with the FEMs. 

## Figures and Tables

**Figure 1 sensors-23-04387-f001:**
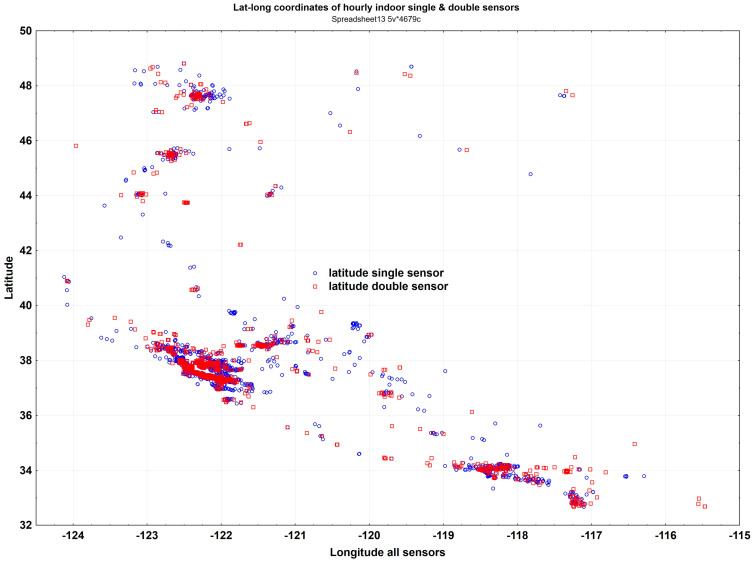
Locations of 4678 PurpleAir indoor monitors in the three West Coast states. Shown are the 3500 PA-I (single sensor) and 1178 PA-II (double sensor) monitors.

**Figure 2 sensors-23-04387-f002:**
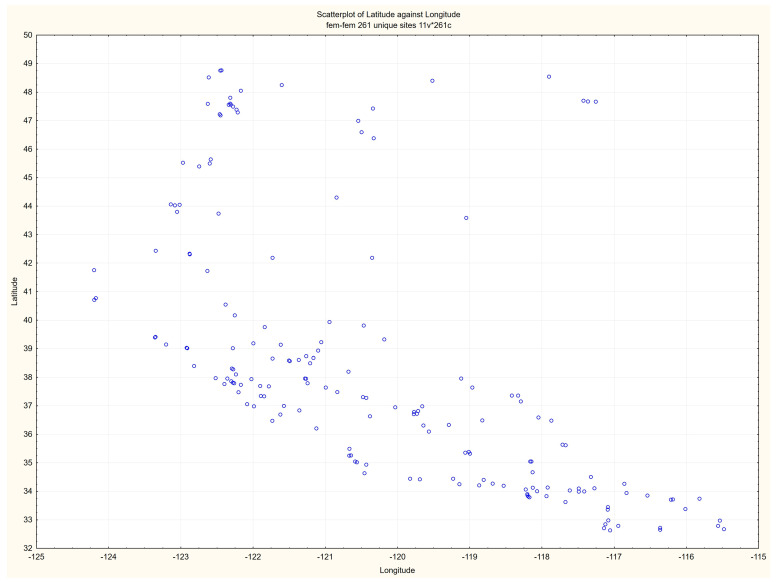
Locations of regulatory sites in the three-state area. There are 261 unique regulatory monitors at 175 sites.

**Figure 3 sensors-23-04387-f003:**
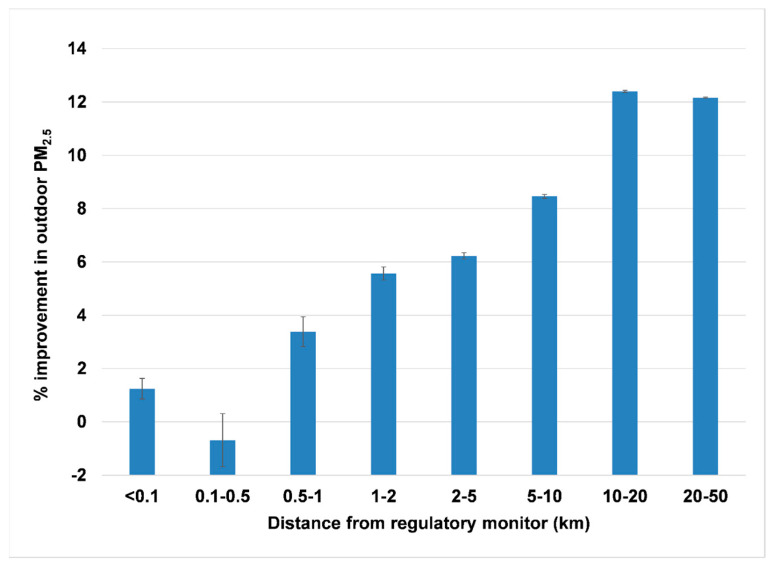
Improvement in outdoor PM_2.5_ with distance from the regulatory monitors.

**Figure 4 sensors-23-04387-f004:**
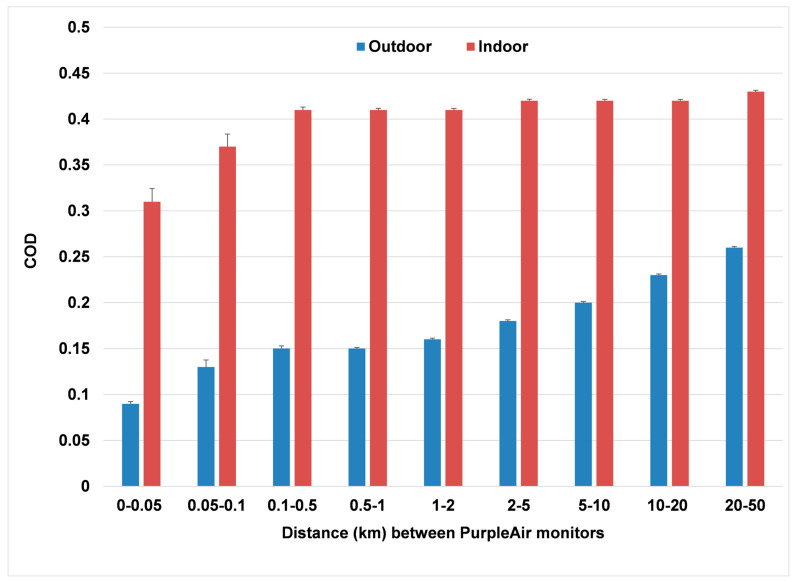
COD as a function of distance apart for outdoor PurpleAir monitors (blue) and indoor PurpleAir monitors (red).

**Table 1 sensors-23-04387-t001:** PM_2.5_ concentrations, correlations, and coefficients of divergence as a function of distance from regulatory monitors for outdoor PurpleAir monitors in three West Coast states between 1 January 2017 and 8 September 2021.

Distance Category (km)	# Site Pairs	Mean Days Per Site Pair *	FRM/FEM Mean PM_2.5_ (SE)	PurpleAir Mean PM_2.5_ (SE)	Pearson Corr. Coeff. (r_p_) (SE)	Coeff. of Divergence (COD) (SE)
<0.1	314	338 (21)	11.7 (0.24)	11.5 (0.30)	0.91 (0.006)	0.24 (0.004)
0.1 to 0.5	42	247 (35)	10.6 (0.64)	10.7 (0.76)	0.90 (0.018	0.24 (0.014)
0.5 to 1	84	316 (28)	9.6 (0.34)	9.3 (0.45)	0.87 (0.014)	0.26(0.008)
1 to 2	399	313 (13)	10.0 (0.16)	9.4 (0.19)	0.87 (0.007)	0.27 (0.004)
2 to 5	2339	301 (5.0)	10.3 (0.07)	9.7(0.09)	0.87 (0.003	0.27 (0.002)
5 to 10	5291	297 (3.6)	10.4 (0.04)	9.5 (0.06)	0.85 (0.002)	0.28 (0.001)
10 to 20	14,721	301 (2.2)	10.4 (0.02)	9.1 (0.03)	0.81 (0.001)	0.30 (0.001)
20 to 50	59,372	305 (1.0)	10.3 (0.01)	9.0 (0.02)	0.76 (o.001)	0.32 (0.000)

***** Number of days with valid daily average concentrations for both the PurpleAir and regulatory monitors at a given site.

**Table 2 sensors-23-04387-t002:** Correlations and coefficients of divergence as a function of distance apart for outdoor and indoor pairs of PurpleAir monitors in three West Coast states between 1 January 2017 and 8 September 2021.

Distance	N Pairs	Mean				Median			
		r_s_ *	r_p_ *	R^2^	CoD	r_s_	r_p_	R^2^	CoD
** *Outdoor* **									
0–0.05 km	868	0.97	0.97	0.95	0.09	0.99	0.998	0.996	0.07
0.05–0.1 km	139	0.96	0.97	0.94	0.13	0.99	0.99	0.99	0.11
0.1–0.5 km	1616	0.93	0.92	0.85	0.15	0.97	0.99	0.98	0.11
0.5–1 km	4302	0.92	0.93	0.85	0.15	0.96	0.98	0.93	0.15
1–2 km	7740	0.91	0.92	0.84	0.16	0.96	0.98	0.96	0.13
2–5 km	7544	0.90	0.91	0.82	0.18	0.94	0.97	0.94	0.15
5–10 km	7591	0.88	0.89	0.79	0.20	0.92	0.95	0.89	0.18
10–20 km	7527	0.84	0.85	0.72	0.23	0.88	0.91	0.83	0.21
20–50 km	7639	0.79	0.80	0.65	0.26	0.83	0.86	0.74	0.24
** *Indoor* **									
0–0.05 km	308	0.75	0.71	0.50	0.31	0.87	0.78	0.61	0.27
0.05–0.1 km	247	0.61	0.57	0.32	0.37	0.64	0.60	0.36	0.36
0.1–0.5 km	2249	0.55	0.48	0.23	0.41	0.56	0.47	0.22	0.38
0.5–1 km	4933	0.52	0.43	0.19	0.41	0.54	0.42	0.17	0.38
1–2 km	7332	0.51	0.42	0.18	0.41	0.53	0.41	0.17	0.39
2–5 km	7424	0.50	0.41	0.17	0.42	0.52	0.40	0.16	0.39
5–10 km	7537	0.48	0.40	0.16	0.42	0.50	0.38	0.15	0.40
10–20 km	7508	0.47	0.38	0.15	0.42	0.49	0.37	0.14	0.40
20–50 km	7589	0.45	0.37	0.14	0.43	0.47	0.35	0.13	0.40

* r_s_ = Spearman correlation coefficient; r_p_ = Pearson correlation coefficient.

**Table 3 sensors-23-04387-t003:** Correlations and coefficients of divergence as a function of distance apart for regulatory monitors in three West Coast states over 5 years (1 January 2017 to 31 December 2021).

Distance (km)	N Pairs	Mean				Median			
		r_s_ *	r_p_ *	R^2^	CoD	r_s_ *	r_p_ *	R^2^	CoD
0 (collocated)	68	0.97	0.93	0.93	0.14	0.99	0.95	0.97	0.13
2 to 5	40	0.92	0.85	0.84	0.19	0.95	0.87	0.90	0.19
5 to 10	56	0.91	0.86	0.83	0.17	0.95	0.92	0.90	0.14
10 to 20	134	0.88	0.86	0.78	0.18	0.90	0.87	0.82	0.18
20 to 50	614	0.80	0.77	0.64	0.24	0.84	0.79	0.70	0.23
50 to 100	1316	0.64	0.63	0.41	0.30	0.69	0.67	0.48	0.29
100 to 200	3536	0.52	0.52	0.27	0.34	0.54	0.56	0.29	0.33

* r_s_ = Spearman correlation coefficient; r_p_ = Pearson correlation coefficient.

**Table 4 sensors-23-04387-t004:** Comparison of FRM and FEM methods over the 5-year period (2017–2021).

	# Pairs	# Days Per Site	Mean				Median			
			r_s_ *	r_p_ *	R^2^	CoD	r_s_ *	r_p_ *	R^2^	CoD
FRM-FRM	23	223	0.96	0.98	0.96	0.10	0.98	0.99	0.99	0.08
FRM-FEM	36	470	0.92	0.96	0.92	0.16	0.94	0.98	0.96	0.13
FEM-FEM	9	345	0.90	0.91	0.82	0.15	0.94	0.94	0.88	0.11

* r_s_ = Spearman correlation coefficient; r_p_ = Pearson correlation coefficient.

## Data Availability

All raw data publicly available from PurpleAir websites (https://www2.purpleair.com/), (https://api.purpleair.com/). All analyzed data available on request from the corresponding author.

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
