# Peer review of "Spatial Variation of PM2.5 Indoors and Outdoors: Results from 261 Regulatory Monitors Compared to 14,000 Low-Cost Monitors in Three Western States over 4.7 Years"

_sensors, 2023, doi:10.3390/s23094387_

Round 1

Reviewer 1 Report

The manuscript "Spatial variation of PM2.5 indoors and outdoors: results from 261 regulatory monitors compared to 14,000 low-cost monitors in three Western states over 4.7 years" provided information on the comparison of large scale monitoring networks using sensors which can be benefit to the future air quality management, in term of availability of air quality data. However, there are some questions in the analysis that the authors should address.

1) The manuscript does not have line number. It is difficult to provide specific comments. Please add in the next revision.

2) There are different reference format use, for example [2] and superscript 1, 27, etc. Also, the first reference is [2] in this manuscript. 

3) There are spacing problems throughout the manuscript, ex. of "(2 spaces)" monitors produced by different manufacturers [25].

4) Abstract: The spatial variation of indoor PM2.5 indoors -- there are 2 indoor. Please revise.

5) Introduction: As a result scores of articles on ... --> what is this scores? The authors should explain more about this. 

6) Introduction: More information are needed. What types of sensors are used? Where are these sensors? Although the authors refer to previous study, but this manuscript need to provide enough information to the reader by it own.   

7) What are information from Figure 1 that haven't been provided in Table 1? Why the authors need Figure 1?

8) Table 1: FRM FEM --> Is it FRM/FEM?

9) Table 2: Indoor: The authors need to provide information about indoor sensors. What are the types of household using these indoor sensors? For sure, if the sensor was placed in the closed room with air purifier, there will be no relationship among indoor air in the same area. 

10) Figure 2: Similar question. Why the authors need Table 2 since Table 2 give the same informaiton?

11) Section 3.4: This comparison is for the monitoring in what distances?

12) Discussion about indoor (section 4.3). The authors should discuss about the situation of indoor air quality. What are the practices people in the area used indoor? Not only indoor sources, but intrusion and treatment (air purifier) also contribute to indoor air quality.

13) The authors did not discuss about sources of air pollution in the area. It is questioned that outdoor sensors placed close to busy traffic roads (if any) has high correlation with sensors placed in an ambient area without emission sources nearby.   

Reviewer 2 Report

The study is done to identify the spatial variability of indoor and outdoor PM2.5 within three states for a 5-year period is studied using regulatory and low-cost PurpleAir monitors. Though this study is relevent and falls in the interest of global environmetal audiance, it requires major corrections befor the final decision on the manuscript. Clear objective and study area description is missing. Resultrs presented poorly. Spatial varitions are not presented any were in the manuscript. a map may be added dipicting spatial variontion. The comments are marked in manuscript file and attached herewith.

Reviewer 3 Report

This is an interesting manuscript on Spatial variation of PM2.5 indoors and outdoors: results from 261 regulatory monitors compared to 14,000 low-cost monitors in three Western states over 4.7 years. The manuscript contains some interesting measurements but tries to do too many things at once, and at some points more scientific clarity is needed. It can be published in Sensors only if these issues are addressed. More specifically:

Main comments

1. The text should be checked for syntax etc. e.g. abstract: “… The spatial variation of

indoor PM2.5 indoors is …”.

2. The title is too long and descriptive. It could be reduced. For example, as “Spatial variation of PM2.5 indoors and outdoors: comparison of results from regulatory and low-cost monitors in three Western states in USA”.

3. Some notions are not very clear, and by themselves can be misleading. E.g. “There is also a steady improvement in outdoor PM2.5 concentrations with increasing distance from the regulatory monitors”. Why is that? Are the monitors located close to sources? Else, it does not make much sense.

4. Introduction: “Second, the correlations and COD results for pairs of PurpleAir indoor and outdoor monitors are calculated. This provides an indication of the replicability of PurpleAir measurements.”. The authors should better explain this. Indoor and outdoor monitors measure different things. Also, what is the spatial relation of each pair of monitors. More information should be provided here.

5. In my view, the manuscript is looking at two different (although related) issues: (a) spatial variability of outdoor pollution and correlation between two types of monitors-and this is useful for identifying the need and adequacy of low cost monitors to cover greater area spatially, and (b) correlation between indoor and outdoor concentrations. I believe the manuscript should be structured in this way and better and more clearly show the above from the very beginning (and not only from the Methods section). Also, at some point, a third issue/aim is introduced: the comparison of two methodologies (FEM and FRM).

6. The last paragraph of the introduction is confusing and needs to be re-written and expanded (it can be broken into two paragraphs, one on what is existing in literature and one on what exactly this study is doing and its novelty. For example:

a) the reader has the impression that the manuscript is looking only at indoor pollution. The article should clearly explain form the beginning if it is looking at indoor, outdoor pollution, or both, and why/how.

b) the authors note “Only with this assumption can health effects due to actual exposure to particles from all sources be related to ambient particles alone. Yet health effects can be expected from some indoor-generated particles such as those created by smoking (tobacco,

marijuana) and high-temperature cooking”. This is forgetting the serious issue of indoor air pollution at developing countries due to heating and cooking using coal, etc. This issue should be at least referred to here (there are many references in the literature).

7. “…respectively27…” (section 2),“It has been shown1” (section 4.3): the correct format for references should be used.

8. Section 2.1. More information should be provided here. 50 km distance do not provide any useful info if one monitor is placed at curbside and the other at a background site. A table with relevant information would be useful, or at least some more information in the text. Where were the measuremts made? In cities? In Rural areas? Was each pair located at a similar type of place?

Also, a figure with the map of the area/s, preferably with the monitor locations (even only the regulatory ones) is necessary.

9. Some more information on the ALT-CF3 algorithm is needed in the Methods section.

10. Figure 1 and text: why as the distance between the PurpleAir and regulatory monitors increased, the PM2.5 concentration declined (i.e., the air quality improved)? This is related to my previous comment regarding the need for more information on the location of the monotors (eg. Regulatory ones) in relation to cities, industrial sources, etc.

11. “The outdoor sites showed mean and median correlations from 0.88-0.99 up to 10 km distance apart.”. This is not very clear. Do you mean the outdoor and indoor sites? Why should we be interested, especially in this section of the manuscript, on the correlation between outdoor sites only or indoor sites only. All this is not clear and should be explained. Its logic, its contribution to the overall aim, etc. For example, for indoor sites what is the meaning/need to look at such correlation? One would expect to see the correlations between outdoor and indoor monitors, i.e. for pairs located in proximity.

12. What is the difference between sections 3.3 and 3.4? Section 3.3 is not very clear (its content and purpose) and should be further explained.

13. The discussion section 4 should be enhanced to clearly show the aim and added value of this research in relation to existing literature.

(a) For example in section 4.1. we see for the first time some information that should exist earlier. Information on separation and vicinity to major sources should be provided earlier. Here, we would like to see some more insight on how separation effects correlation and COD, as well as some proposals based on the many measurements of this study on limits of distances between monitors, on the density of the monitors network to have a good overview/representation of an area, etc, which would optimize network densities and combined use of regulatory and low-cost monitors. Focused additional use of other literature should be very useful as well.

(b) In 4.2, more information should be provided on the reasoning for the comparison, the type of locations of monitors that were correlated between them (e.g. both urban background, traffic oriented and urban background, etc…). This would make easier the interpretation of results and the optimization of the design/density of such monitoring networks.

(c) Section 4.2 and 4.3: one would expect to see here intercomparison between indoor and outdoor measurements. This is missing and should be added. It should be also linked to what is argued in section 4.3.

14. Conclusions: this section needs to be enhanced based on the comments above.

15. The results shown in section 4.3 alone do not show that “We conclude that indoor PM2.5 exposures cannot be estimated quantitatively from outdoor measurements.”. I believe that this generalization is a bit weak/problematic. Correlation of outdoor and indoor measurements are at least necessary in this respect. And the framework of each study is also relevant of course.

Reviewer 4 Report

Manuscript ID-Sensors-2333973

In this manuscript (MS) by Lance Wallace and Tongke Zhao, spatial variation of PM2.5 indoors and outdoors was examined in three Western states over 4.7 years, January 2017 and September 2021. Indoor and outdoor PM2.5 is studied using regulatory and low-cost PurpleAir monitors. About 260 regulatory monitors and ~10,000 outdoor and ~4,000 indoor PurpleAir monitors are included. Firstly, authors compare the correlations and CODs between PurpleAir monitors and regulatory sites using Federal Equivalent Methods (FEM) or Federal Reference Methods (FRM). Second, the correlations and COD results for pairs of PurpleAir indoor and outdoor monitors are calculated. Lastly, the correlations and CODs of pairs of PurpleAir monitors at a range of distances from 100 m to 50 km are measured.

This is an interesting study. The authors did a good job clearly explaining the method and task of the work and the presentation of the data was well done. In my opinion, this is a good work and it should be published, but before of that I suggest some minor revisions.

Comments and Suggestions

1.      Abstract section: In my opinion, I think authors should remove [1] to Introduction section. References number here, I think that it is an unsuitable.

2.      Introduction section: Introduction is suitable.

3.      Methods and Materials: Please check and revise superscript, “September 8, 20211” and “respectively27 I think it is an unsuitable.

4.      Methods and Materials: The authors mentioned in abstract section that most of data were collected in an earlier study. However, why in abstract mentioned “260 regulatory monitors” while the methods and materials “~160 regulatory sites” Is it correct number? what is the reason? In order not to mislead the potential readers. Please, explain the reason.

5.      Methods and Materials: Why did authors set the distance range from 100 m to 50 km? Please specify the background or criteria of this distance category.

6.      Figure 1: The author indicated the PM2.5 concentration declined (i.e., the air quality improved) if the distance between the PurpleAir and regulatory monitors are increased. However, when we consider to bar graph at distances from 20-50 km appear lower % improvement in outdoor PM2.5 than 10-20 km distances. Could you please explain the reason?

7.      4.1. Comparisons of PurpleAir with regulatory sites: Please follow journal format “algorithm1,26-28

8.      In section of Intercomparisons of PurpleAir outdoor sites: Why the results appear robust against extreme variations in PM2.5 concentrations? Which seasonal or years are the most extreme variations in PM2.5 concentrations?

9.      4.3.Comparisons of PurpleAir indoor sites section: Please follow journal format “shown1

10.   Table 4 and 4.4. Intercomparison of regulatory sites: Dose it means that PurpleAir PM2.5 measurements can agree well with result of FEM-FEM pairs and can validate the algorithm and calibration factor for all PurpleAir monitors?

Reviewer 5 Report

1.       It is not customary to use reference in the abstract, it is better to use the general data of the study instead of referencing.

2.       Many sections of the introduction lack references, it is necessary to specify and add references to pages 2 and 3 in the introduction section.

3.       Although a comparison has been made between these regulatory monitors and low-cost monitors, their accuracy has not been mentioned much

4.       It seems that two referencing styles have been used in the introduction and discussion

Round 2

Reviewer 3 Report

Line 294: it is “2017” and not “2917”

Lines 360-361: “We conclude that indoor PM2.5 exposures cannot be 360 estimated quantitatively from outdoor measurements alone.”. It should be rephrased to “We conclude that, in the context of the present study, indoor PM2.5 exposures cannot be 360 estimated quantitatively from outdoor measurements alone.”.

Author Response

Line 294: it is “2017” and not “2917”

CORRECTED

Lines 360-361: “We conclude that indoor PM2.5 exposures cannot be 360 estimated quantitatively from outdoor measurements alone.”. It should be rephrased to “We conclude that, in the context of the present study, indoor PM2.5 exposures cannot be 360 estimated quantitatively from outdoor measurements alone.”.

AGREED--CHANGE MADE